# Bacterial Community Structure in Rhizosphere of Barley at Maturity Stage

**Siyu Zhang** [1,†], **Yue An** [1,†], **Yu Zhou** [1], **Xiaofang Wang** [1], **Yiqing Tang** [2], **Daorong Zhang** [2], **Genlou Sun** [3], **Qifei Wang** [4] and **Xifeng Ren** [1,*]

1   Hubei Hongshan Laboratory, College of Plant Science and Technology, Huazhong Agricultural University, Wuhan 430070, China; 2021301110109@webmail.hazu.edu.cn (S.Z.); hzauay@163.com (Y.A.); zhouyu@mail.hazu.edu.cn (Y.Z.); wangxiaofangchina@163.com (X.W.)
2   Xiangyang Academy of Agricultural Sciences, Xiangyang 441057, China; xftqy811006@163.com (Y.T.); dongfanghong1998@163.com (D.Z.)
3   Department of Biology, Saint Mary's University, Halifax, NS B3H 3C3, Canada; genlou.sun@smu.ca
4   Institute of Crop and Nuclear Technology Utilization, Zhejiang Academy of Agricultural Sciences, Hangzhou 310021, China; fei920214@sina.cn
*   Correspondence: renxifeng@mail.hzau.edu.cn
†   These authors contributed equally to this work.

**Abstract:** The crop rhizosphere is the main site of soil microbial activities. Understanding the structure and diversity of microbial communities in the crop rhizosphere will help us reveal interactions between rhizosphere microorganisms and plant growth. In this study, the rhizosphere soil was collected from 35 cultivated barley varieties at the mature stage. To investigate the structure and diversity of bacterial communities in the rhizosphere of different barley varieties, the 16S rDNA gene of microorganisms from the soil was sequenced using Illumina MiSeq next-generation high-throughput sequencing technology. The results showed that 13, 25, 49, and 59 bacterial flora with relative abundance >1% were detected from 35 barley rhizosphere samples at the phylum, class, order, and family levels, respectively. The abundance of bacteria among varieties differed relatively little, but the abundance of the same bacteria in rhizospheres of different varieties was different. In addition, both the cluster analysis and principal component analysis (PCA) divided the 35 samples into three clusters at the phylum level. Groups III and IV showed significantly higher abundance than group II in Proteobacteria, while group II exhibited significantly higher abundance of Chloroflexi than groups III and IV. This finding provides a realistic basis for further using the relationship between barley rhizosphere microorganisms and barley growth to improve the resistance and quality of barley.

**Keywords:** barley; maturity period; rhizosphere; bacterial community; structural characteristics

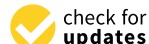



## 1. Introduction

The rhizosphere refers to the microenvironment that is different from soil in physical, chemical, and biological properties due to the activity of plant roots [1]. In this special ecological environment, the number, species, and metabolic activities of soil microorganisms are several or even dozens of times greater than in non-rhizosphere areas, which is an important place for material exchange between plants and soil [2,3]. Nutrients and microorganisms in soil can change the physical and chemical properties of soil and provide nutrients for plant growth [4]. The secretions produced by plant roots also affect the structure and composition of microbial communities around the rhizosphere [5]. Therefore, the study of crop rhizosphere microbial community structure and diversity is helpful for us to understand the interaction between rhizosphere microorganisms and plant growth, and the mechanism of different varieties affecting rhizosphere microbial community, thus making it possible to use rhizosphere-beneficial microorganisms to improve crop varieties [6].

Barley has more than 5000 years of cultivation history in China [7]. It is mainly used as feed and beer industrial raw materials, and it is also the most important rotation crop and

functional health food for Tibetan compatriots [8]. In recent years, with the improvement of people's living standards and the rapid development of the processing industry, the beer and feeding value of barley has been further improved [9]. The development of barley production is not only of great significance to the rapid development of the national economy, but also plays an important role in helping the poverty-stricken areas in Tibet to get rid of poverty. Meanwhile, barley has the characteristics of wide adaptability, strong resistance to stress, and short growth period [10]. It is widely cultivated in northeast China, northwest China, south China, Huanghuai, Yunnan-Guizhou, and the Qinghai-Tibet Plateau [8]. Therefore, whether barley has a wide range of adaptability, salt and alkali resistance, barren resistance, and other stress resistance is related to its rhizosphere micro-ecological environment, and determining the characteristics of the composition of these microbial communities is an urgent matter in terms of barley's wide adaptability and stress resistance [11,12]. Studies on the effects of soil on barley growth and development and barley rhizosphere microbial community structure are limited, which seriously affects the further development of the barley industry. Next-generation sequencing has become an effective tool for characterizing the soil rhizosphere microbiome [13], which can reveal the diversity and complexity of microbial communities' in situ environment. At present, some progress has been made in the study of the microbial diversity community structure of crops, such as *Arabidopsis thaliana*, rice, maize, and potato, using high-throughput sequencing [14–17]. However, little is known about the effects of different crop varieties on rhizosphere microbial activities and community compositions [18,19]. Therefore, we examined the rhizosphere bacterial community structure in the mature stage of 35 barley varieties. The goal was to determine the dominant bacterial communities present in the rhizosphere of barley, as well as to assess the diversity and differences among the different varieties. The results provide knowledge for a comprehensive understanding of the interaction between barley roots and the microbial community.

## 2. Materials and Methods

### 2.1. Plant Growth Conditions

The 35 cultivated barley varieties were provided by the Triticeae research group of Huazhong Agricultural University (Table 1). All materials were planted in the experimental field of Huazhong Agricultural University. Each variety was planted in 3 rows with 1.5 m row length, 20 cm row spacing, and 3 repetitions. The field management was consistent with regular field cultivation. The experimental field (longitude 114.34, latitude 30.47, average altitude 17 m) belongs to the subtropical monsoon climate. The annual precipitation is 1998 mm. Before sowing, we randomly collected soil samples from ten sites (three soil samples per site) in the experimental plot to test their fertility using seven chemical properties (Table 2). The fertility of the experimental plot before sowing was uniform, and the average level of the seven soil chemical properties are shown in Table 2.

**Table 1.** The 35 cultivated barley varieties this study used.

| Code | Variety Name | Code | Variety Name | Code | Variety Name |
|------|-------------|------|-------------|------|-------------|
| R1 | Ganpi No.2 | R13 | Yunpi No.2 | R25 | Edamai 820352 |
| R2 | Ganpi No.3 | R14 | Yunnan S-500 | R26 | Huadamai 1707 |
| R3 | Ganpi No.5 | R15 | Huadamai No.2 | R27 | Huadamai 16316 |
| R4 | Kenpimai No.1 | R16 | Huadamai No.7 | R28 | Huadamai No.13 |
| R5 | Kenpimai No.2 | R17 | Huadamai No.9 | R29 | Huadamai No.15 |
| R6 | Kenpimai No.4 | R18 | Suhua No.2 | R30 | Huadamai No.18 |
| R7 | Kenpimai No.6 | R19 | Kanghanluodamai | R31 | Huadamai No.20 |
| R8 | Kenpimai No.8 | R20 | Hailaerdamai | R32 | Edamai 523898 |
| R9 | Kenpimai No.9 | R21 | Huadamai1539 | R33 | Edamai 720135 |
| R10 | Kenpimai No.10 | R22 | Changfupi No.1 | R34 | Edamai 720033 |
| R11 | Kenjianpi No.3 | R23 | Edamai 522600 | R35 | Huadamai 16312 |
| R12 | Edamai 029 | R24 | Edamai 730135 | | |

**Table 2.** The average level of the seven soil chemical properties of the experimental plot before sowing.

| PH | Ammonium Nitrogen (mg/kg) | Nitrate Nitrogen (mg/kg) | Organic Carbon (‰) | Available Phosphorus (mg/kg) | Available Potassium (mg/kg) | Effective Boron (mg/kg) |
|---|---|---|---|---|---|---|
| 6.3 | 26.92 | 99.78 | 19.92 | 48.57 | 149.4 | 0.12 |

*2.2. Collection of Soil Samples*

The rhizosphere soil of 35 barley varieties was collected at the maturity stage of the barley, with 3 replicates for each sample. During the collection, 2 cm of surface soil of the barley roots was removed, and the soil profile of 2–15 cm depth was taken. Then, the barley plants were gently dug up and pulled out, and the soil attached to the roots was gently shaken off. The soil closely attached to the root surface was the rhizosphere soil. We took three samples from each replicate to obtain 9 soil samples for each variety. Then we collected rhizosphere soil from these 9 samples of each variety. Briefly, the roots with fine roots of 2–7 cm depth were cut for obtaining the rhizosphere soil samples. Barley roots of 8–15 cm depth were not considered because they are not sufficient for the collection of rhizosphere soil samples. Moreover, they might show significant heterogeneity with roots of 2–7 cm depth in the structure and diversity of the bacterial community. Fine roots of 2–7 cm depth were cut, and the remaining soil was brushed with a small brush into a numbered sterile self-sealing bag, placed in an ice box, and brought back to the laboratory. In order to ensure sufficient rhizosphere soil for sequencing, we mixed the 9 soil samples from each variety.

*2.3. Rhizosphere Bacterial Genome Sequencing*

A soil DNA extraction kit (E.Z.N.A.Soil DNA-Kit) was used to extract total microbial DNA according to the manufacturer's instructions. The extracted DNA was dissolved in 50 μL sterile ultrapure water, mixed evenly, and stored at −80 °C.

The 16S rDNA of each sample was amplified by PCR. The PCR products were detected by 2% agarose gel, purified by AMPure XT beads (Beckman Coulter Genomics, Danvers, MA, USA), and quantified by Qubit (lnvitrogen, Carlsbad, CA, USA). We assessed the size and concentration of each amplified library with the Agilent 2100 Bioanalyzer (Agilent, Santa Clara, CA, USA). Finally, the libraries were sequenced based on Illumina Miseq platform by Shanghai Majorbio Company (Shanghai, China).

*2.4. Bioinformatical Analyses*

After sequencing, Length Adjustment of Short Reads (FLASH) [20] and Trimmomatic v.0.38 [21] software were used to filter the sequences, and the sequences with similarities over 97% were grouped into an Operational Taxonomic Unit (OTU) for taxonomic analysis. Ribosomal Database Project (RDP) classifier software was used to annotate the OTU sequence and analyze the community composition of the samples at different classification levels (http://rdp.cme.msu.edu/classifier/classifier.jsp; accessed on 21 July 2023) [22]. Mothur v.1.11.0 software was used to analyze the difference of α-diversity between groups [23]. UniFrac metric software (v.1.9.0) was used for β-diversity sample clustering and for PCA (https://www.majorbio.com/; accessed on 21 July 2023). Pearson's correlation analysis was performed using the R package corrplot (v.0.92) [24].

**3. Results**

*3.1. Quality Analysis of Barley Rhizosphere Bacteria Sequencing*

Through high-throughput sequencing of 35 rhizosphere soil samples from 35 barley varieties, we obtained 1,411,004 valid sequences with an average length of 416 bp (Table S1). The dilution curves of 35 rhizosphere soil samples tended to be flat, indicating that the sequencing data were reasonable and the sequencing depth was sufficient, and the results

obtained could better reflect the bacterial community composition structure of each variety (Figure 1). At the 97% similarity level, we detected a total of 5095 OTUs, belonging to 35 phyla, 117 classes, 267 orders, and 429 families.

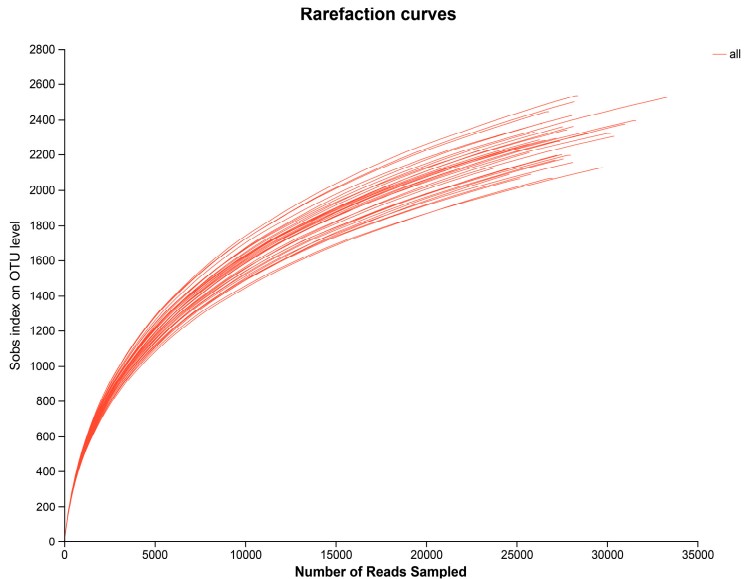

**Figure 1.** Dilution curves of 35 barley rhizosphere soil samples.

### 3.2. Bacterial Community Composition in Barley Rhizosphere at Phylum Level

A total of 35 bacterial phyla were detected in the 35 rhizosphere soil samples collected from barley varieties at the mature stage (Figure 2). Among them, 25 phyla were shared by 35 soil samples. The number of dominant phyla (relative abundance > 1%) was 13 (Actinobacteriota, Proteobacteria, Acidobacteriota, Chloroflexi, Gemmatimonadota, Firmicutes, Bacteroidota, Myxococcota, Verrucomicrobiota, Planctomycetota, Methylomirabilota, Nitrospirota, Patescibacteria). The five phyla with relatively high abundance were Actinobacteriota (19.01–31.93%), Proteobacteria (15.56–35.84%), Acidobacteriota (5.47–26.85%), Chloroflexi (9.21–17.83%), and Gemmatimonadota (3.01–8.03%). In the 35 samples, R33 (Edamai 720135) showed relatively high abundance of Proteobacteria, while R32 (Edamai 523898) showed the lowest abundance of Actinobacteriota (Figure 2).

Among the 13 dominant phyla, Actinobacteriota had the highest average abundance (26.39%) and the lowest coefficient of variation (12.38%) (Table 3), suggesting that this phylum not only had the highest abundance in barley, but also was relatively stable among different varieties. The abundance of Proteobacteria, Acidobacteriota, and Chloroflexi in 35 different barley varieties was higher, at 23.02%, 14.89%, and 13.28%, respectively. Proteobacteria was the dominant phylum, second only to Actinobacteriota, and its abundance varied greatly among 35 rhizosphere soil samples from different barley varieties, ranging from 15.56% to 35.84%. The abundance of Acidobacteriota was also significantly different among rhizosphere soil samples, ranging from 5.47% to 26.85%. Chloroflexi was also relatively stable in 35 rhizosphere soil samples, with an average abundance of 13.28%. The average abundance of the other 9 dominant phyla was less than 10%, but the coefficient of variation of Firmicutes, Bacteroidota, Verrucomicrobiota, Planctomycetota, Methylomirabilota, and Patescibacteria reached more than 36%. Among them, the bacterial phylum with the largest coefficient of variation was Planctomycetota, accounting for 52.52%. The result indicated that the rhizosphere soil of 35 different barley varieties had a wide range of variation in these six bacterial phyla with an average abundance of less than 10%.

In addition, except 25 common phyla, some bacterial communities were low in abundance and only contained in some samples, such as WS2 (0.004–0.073%), Deinococcota (0.003–0.40%), Elusimicrobiota (0.003–0.062%), Fibrobacterota (0.003–0.215%), SAR324_cladeMarine_group_B (0003–0.033%), Spirochaetota (0.003–0.032%), GAL15 (0.003–0.039%), WPS-2 (0.004–0.025%),

Abditibacteriota (0.004–0.018%), and Dadabacteria (0.013–0.011%) (Figure S1). Dadabacteria only existed in the rhizosphere soil samples from two barley varieties, being the least covered sample among all bacteria and belonging to a special bacterial phylum. Some bacterial phyla that have a certain effect on plant growth reported in previous studies were also observed in lower abundance, among which the more typical was Deinococcus-Thermus (Deinococcota) (Figure S1), detected in 14 of the 35 samples in our study (Figure S1).

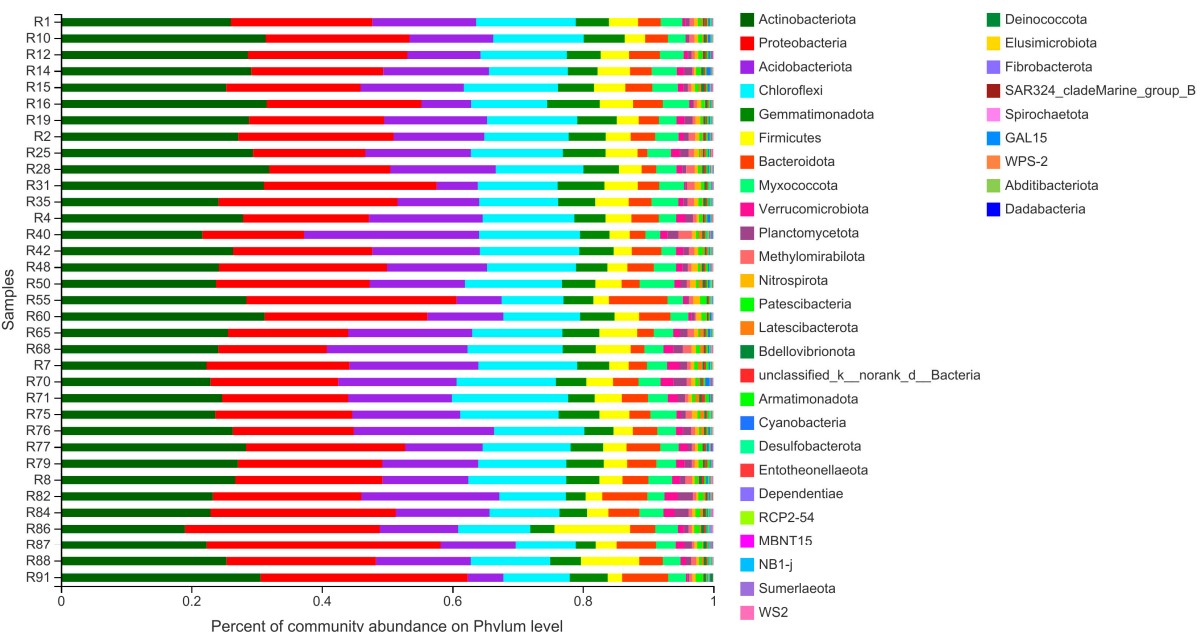

**Figure 2.** Abundance at the phylum level of barley rhizosphere bacteria.

**Table 3.** Phenotypic variation in 13 dominant phyla.

| Phylum Name | Range | Minimum | Maximum | Median | Average | Standard Deviation | Coefficient of Variation |
|---|---|---|---|---|---|---|---|
| Actinobacteriota | 0.1292 | 0.1901 | 0.3193 | 0.2635 | 0.2639 | 0.0327 | 0.1238 |
| Proteobacteria | 0.2028 | 0.1556 | 0.3584 | 0.2212 | 0.2302 | 0.0458 | 0.1988 |
| Acidobacteriota | 0.2138 | 0.0547 | 0.2685 | 0.1531 | 0.1489 | 0.0450 | 0.3025 |
| Chloroflexi | 0.0862 | 0.0921 | 0.1783 | 0.1365 | 0.1328 | 0.0193 | 0.1455 |
| Gemmatimonadota | 0.0502 | 0.0301 | 0.0803 | 0.0510 | 0.0517 | 0.0101 | 0.1959 |
| Firmicutes | 0.0938 | 0.0221 | 0.1159 | 0.0382 | 0.0423 | 0.0179 | 0.4219 |
| Bacteroidota | 0.0747 | 0.0152 | 0.0899 | 0.0386 | 0.0403 | 0.0148 | 0.3678 |
| Myxococcota | 0.0306 | 0.0224 | 0.0530 | 0.0309 | 0.0324 | 0.0064 | 0.1967 |
| Verrucomicrobiota | 0.0183 | 0.0016 | 0.0199 | 0.0098 | 0.0103 | 0.0048 | 0.4700 |
| Planctomycetota | 0.0213 | 0.0024 | 0.0237 | 0.0085 | 0.0098 | 0.0051 | 0.5252 |
| Methylomirabilota | 0.0185 | 0.0023 | 0.0208 | 0.0057 | 0.0071 | 0.0036 | 0.5026 |
| Nitrospirota | 0.0075 | 0.0032 | 0.0107 | 0.0062 | 0.0063 | 0.0019 | 0.3038 |
| Patescibacteria | 0.0088 | 0.0020 | 0.0108 | 0.0049 | 0.0053 | 0.0019 | 0.3652 |

### 3.3. Community Composition of Barley Rhizosphere Bacterial Community at Class Level

At the class level of bacterial classification, 117 different bacterial classes were detected in 35 rhizosphere soil samples. There were 25 dominant classes with relative abundance > 1% and 92 minor classes with relative abundance < 1%. The dominant classes determined in each sample were completely consistent (Figure 3). Among the twenty-five dominant classes, there were seven classes with average relative abundance > 5%, such as Actinobacteriota (10.49–19.36%), Alphaproteobacteria (8.61–21.75%), Gammaproteobacteriota (6.53–14.09%), Thermoleophilia (4.49–14.06%), Vicinamibacteria (1.52–17.41%), Chloroflexia (3.76–8.94%), and Gemmatimonadetes (2.97–7.95%) (Table 4).

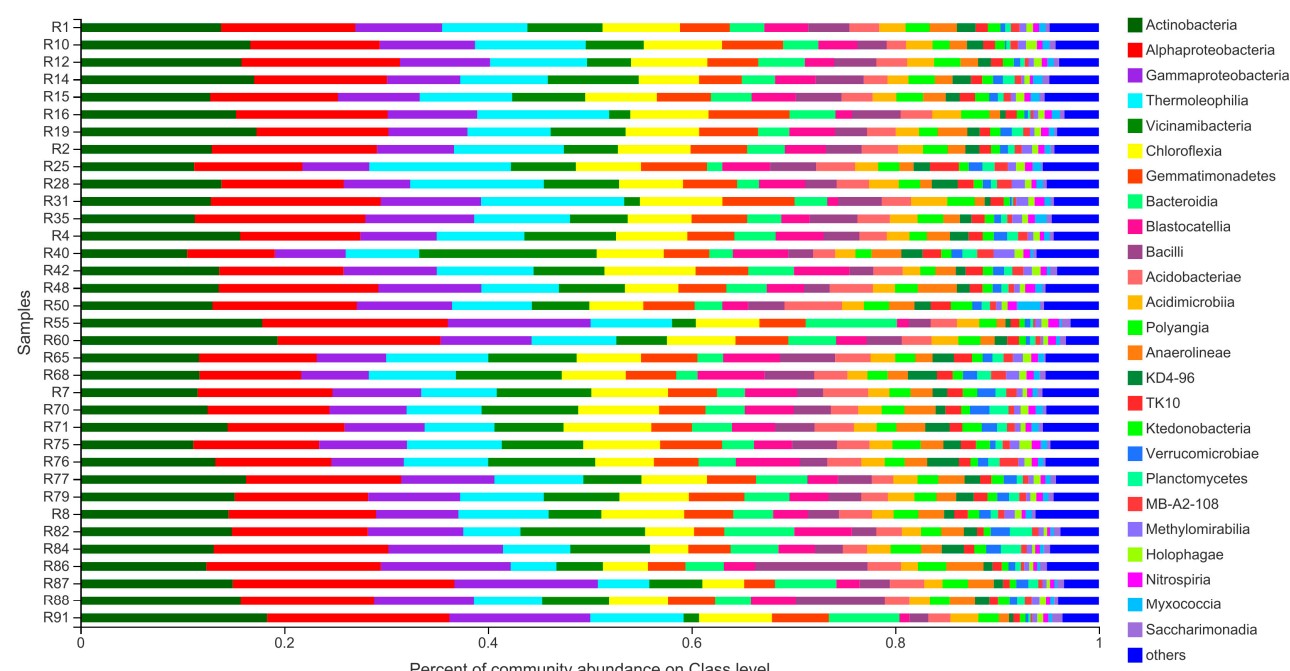

**Figure 3.** Abundance at the class level of barley rhizosphere bacteria.

**Table 4.** Phenotypic variation in 25 dominant classes.

| Class Name | Range | Minimum | Maximum | Median | Average | Standard Deviation | Coefficient of Variation |
|---|---|---|---|---|---|---|---|
| Actinobacteria | 0.0887 | 0.1049 | 0.1936 | 0.1707 | 0.1416 | 0.0226 | 0.1596 |
| Alphaproteobacteria | 0.1314 | 0.0861 | 0.2175 | 0.1305 | 0.1397 | 0.0270 | 0.1932 |
| Gammaproteobacteria | 0.0756 | 0.0653 | 0.1409 | 0.0718 | 0.0904 | 0.0206 | 0.2277 |
| Thermoleophilia | 0.0957 | 0.0449 | 0.1406 | 0.0860 | 0.0877 | 0.0222 | 0.2532 |
| Vicinamibacteria | 0.1589 | 0.0152 | 0.1741 | 0.0893 | 0.0690 | 0.0309 | 0.4478 |
| Chloroflexia | 0.0518 | 0.0376 | 0.0894 | 0.0591 | 0.0664 | 0.0123 | 0.1859 |
| Gemmatimonadetes | 0.0498 | 0.0297 | 0.0795 | 0.0421 | 0.0504 | 0.0099 | 0.1967 |
| Bacteroidia | 0.0745 | 0.0150 | 0.0895 | 0.0326 | 0.0396 | 0.0148 | 0.3744 |
| Bacilli | 0.0915 | 0.0189 | 0.1104 | 0.0474 | 0.0386 | 0.0177 | 0.4586 |
| Blastocatellia | 0.0563 | 0.0097 | 0.0660 | 0.0397 | 0.0386 | 0.0141 | 0.3651 |
| Acidobacteriae | 0.0375 | 0.0190 | 0.0565 | 0.0233 | 0.0306 | 0.0073 | 0.2395 |
| Acidimicrobiia | 0.0188 | 0.0168 | 0.0356 | 0.0207 | 0.0233 | 0.0038 | 0.1613 |
| Polyangia | 0.0149 | 0.0148 | 0.0297 | 0.0258 | 0.0220 | 0.0038 | 0.1748 |
| Anaerolineae | 0.0298 | 0.0086 | 0.0384 | 0.0174 | 0.0212 | 0.0072 | 0.3396 |
| KD4-96 | 0.0269 | 0.0041 | 0.0310 | 0.0181 | 0.0149 | 0.0064 | 0.4293 |
| TK10 | 0.0221 | 0.0064 | 0.0285 | 0.0111 | 0.0130 | 0.0042 | 0.3257 |
| Ktedonobacteria | 0.0156 | 0.0054 | 0.0210 | 0.0106 | 0.0102 | 0.0034 | 0.3344 |
| Verrucomicrobiae | 0.0179 | 0.0012 | 0.0191 | 0.0097 | 0.0095 | 0.0046 | 0.4809 |
| Planctomycetes | 0.0202 | 0.0016 | 0.0218 | 0.0117 | 0.0087 | 0.0048 | 0.5488 |
| MB-A2-108 | 0.0167 | 0.0019 | 0.0186 | 0.0066 | 0.0075 | 0.0040 | 0.5345 |
| Methylomirabilia | 0.0185 | 0.0023 | 0.0208 | 0.0029 | 0.0071 | 0.0036 | 0.5040 |
| Holophagae | 0.0093 | 0.0028 | 0.0121 | 0.0054 | 0.0065 | 0.0019 | 0.2993 |
| Nitrospiria | 0.0075 | 0.0032 | 0.0107 | 0.0032 | 0.0062 | 0.0019 | 0.3027 |
| Myxococcia | 0.0206 | 0.0023 | 0.0229 | 0.0094 | 0.0061 | 0.0036 | 0.5866 |
| Saccharimonadia | 0.0087 | 0.0016 | 0.0103 | 0.0067 | 0.0048 | 0.0019 | 0.3924 |

Among the twenty-five dominant classes, Actinobacteriota and Alphaproteobacteria were the only two dominant classes with an average relative abundance > 10% in the 35 barley rhizosphere soil samples, the average relative abundance of which was 14.16% and 13.97%, and the coefficient of variation was 15.96% and 19.32%, respectively (Table 4). There were fifteen dominant classes with an average relative abundance of 1–10%, ten

with a coefficient of variation of 16–35%, and five with a coefficient of variation of >36%. There were eight dominant classes with an average relative abundance of <1%, and two classes with a coefficient of variation between 16% and 35%. There were six classes with a coefficient of variation >36%. The coefficients of variation of all 25 dominant classes were greater than 15%. Among them, Actinobacteriota, with the largest abundance, had the smallest coefficient of variation (15.96%), and Myxococcia, with the smallest abundance, had the largest coefficient of variation (58.66%) (Table 4).

*3.4. The Bacterial Community Composition of Barley Rhizosphere at the Level of Order and Family*

At the order level of bacterial classification, a total of 49 dominant bacterial orders (relative abundance > 1%) were detected in the 35 rhizosphere soil samples (Figure S2). The top 10 dominant orders were Vicinamibacteria (1.46–16.56%), Rhizobiales (4.61–9.81%), Burkholderiales (4.79–11.43%), Chloroflexales (2.15–7.72%), Gaiellales (2.34–8.96%), Gemmatimonadales (2.97–7.96%), Micrococcales (1.61–8.98%), Solirubrobacterales (2.07–5.50%), Sphingomonadales (1.28–4.82%), and Bacillales (1.64–9.94%). Among the top ten dominant bacterial orders, there were three orders with an average abundance of >6%, namely Vicinamibacteria (6.80%), Rhizobiales (6.82%), and Burkholderiales (6.44%), and their coefficients of variation were 44.14%, 22.89%, and 20.91%, respectively. Among the seven orders with an average abundance of less than 6%, Gemmatimonadales (5.04%) was the flora with the smallest coefficient of variation (19.67%) in the top 10 orders, indicating that it was more stable than others. Although the average abundance of Bacillales was 3.27%, the coefficient of variation was 50.98%. Among the top 10 orders in abundance, the coefficient of variation was greater than 16%, indicating that the rhizosphere bacteria of 35 barley varieties had extensive variation at the order level (Table S2).

A total of 59 dominant families (relative abundance > 1%) were detected in the 35 rhizosphere soil samples from barley at the mature stage (Figure S3). The top 10 dominant families were Roseiflexaceae (2.07–7.41%), Gemmatimonadaceae (2.97–7.95%), f_norank_o_Vicinamibacterales (0.84–9.15%), Xanthobacteraceae (2.53–5.40%), f_norank_o_Gaiellales (1.47–6.04%), Sphingomonadaceae (1.28–4.82%), Bacillaceae (1.58–9.91%), Vicinamibacteraceae (0.61–7.41%), Chitinophagaceae (0.95–5.73%), and Pyrinomonadaceae (0.59–4.52%). Among the top ten dominant families, only two families with an average abundance > 5% were found: Roseiflexaceae (5.19%) and Gemmatimonadaceae (5.04%), with a coefficient of variation of 23.44% and 19.67%, respectively. Among them, Gemmatimonadaceae is the bacterial group with the largest abundance and the smallest coefficient of variation, indicating that this flora is relatively stable among samples from different varieties. Chitinophagaceae (2.44%) and Pyrinomonadaceae (2.40%) had a mean abundance of less than 3%, and the coefficients of variation were 31.86% and 41.92%, respectively. The average abundance of the other six families ranged from 3% to 4%. Among the top ten families with the average abundance, four families had a coefficient of variation of >36%, and the coefficient of variation of relative abundance of the remaining six samples was between 16% and 35%, indicating that the rhizosphere bacteria in the rhizosphere soil from 35 barley varieties had a wide range of variation at the family level (Table S3).

*3.5. Correlation Analysis of Dominant Bacterial Community in Barley Rhizosphere*

The 13 dominant bacteria at the phylum level (relative abundance > 1%) showed a wide range of correlation (Figure 4). Verrucomicrobiota and Planctomycetota had the highest correlation coefficient of 0.85. Proteobacteria and Methylomirabilota, Proteobacteria and Acidobacteriota, Proteobacteria and Chloroflexi, Bacteroidota and Methylomirabilota, Bacteroidota and Acidobacteriota, Bacteroidota and Chloroflexi, and Patescibacteria and Chloroflexi were significantly negatively correlated ($p < 0.01$). Proteobacteria and Acidobacteriota had the highest negative correlation (−0.66). These results suggested that these highly correlated bacterial communities might have cooperative (positive correlation) or adversarial relationships (negative correlation) with each other.

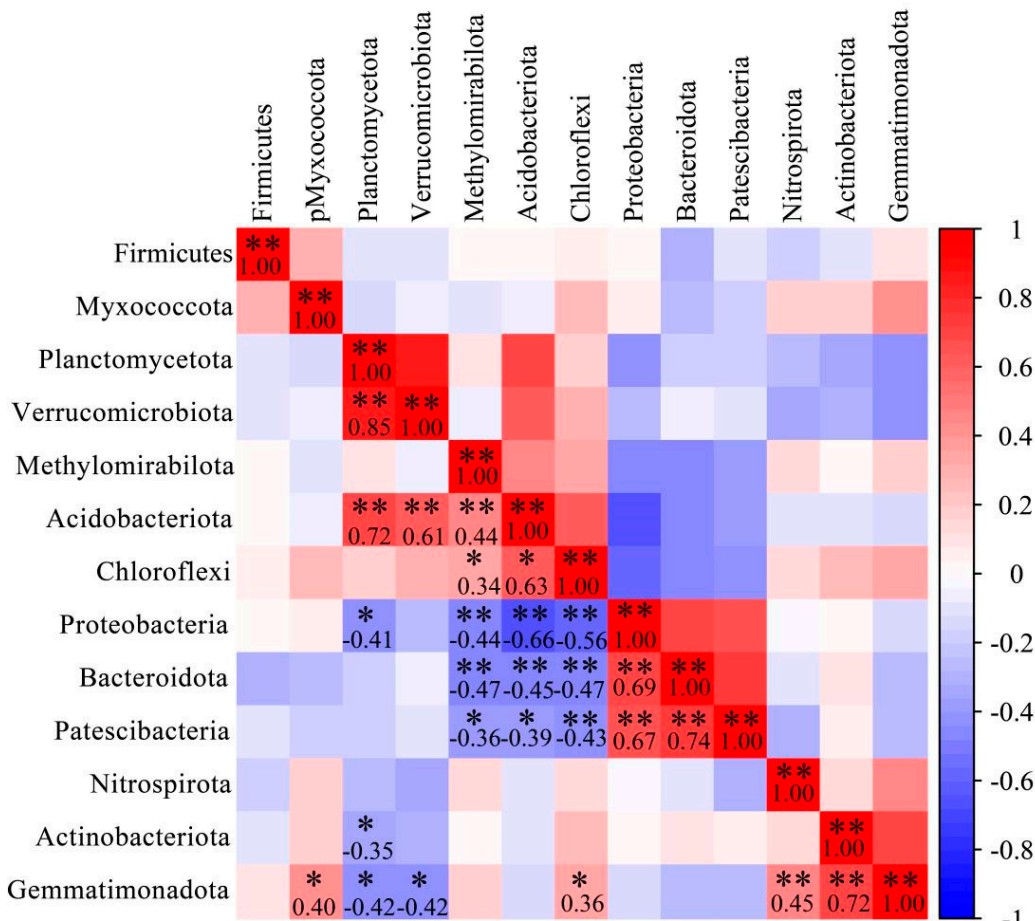

**Figure 4.** Correlation of dominant bacterial communities in barley rhizosphere at phylum level. * and ** indicate significant correlation at 0.05 and 0.01 levels, respectively.

The correlation among 25 dominant classes (relative abundance > 1%) of bacteria from 35 barley rhizosphere soils is shown in Figure 5. Among them, Thermoleophilia and Gemmatimonadetes, Verrucomicrobiae and Planctomycetes, Gammaproteobacteria and Alphaproteobacteria, and Vicinamibacteria and Blastocatellia showed significantly positive correlation ($p < 0.01$), and the correlation coefficients were 0.86, 0.84, 0.83, and 0.83, respectively. Alphaproteobacteria was negatively correlated with Blastocatellia, MB.A2.108, and Vicinamibacteria, and the correlation coefficients were −0.69, −0.69, and −0.64, respectively ($p < 0.01$).

*3.6. Alpha Diversity of Bacterial Community Structure in Barley Rhizosphere*

Alpha diversity analysis is a comprehensive index reflecting the diversity, richness, and evenness of species in the ecosystem. Among them, sobs, chao, and ace are indicators reflecting community richness, while Shannon and Simpson are indicators reflecting community diversity, and coverage reflects community coverage. Table 5 shows that the minimum coverage is 97.35%, indicating that the sequencing depth is sufficient, and the sobs index is stable, around 2273. Among the five indexes reflecting community richness and diversity, notwithstanding that the coefficient of variation of the Simpson index of 0.1925 was greater than 0.1, the coefficient of variation of other indexes was between 0.01 and 0.05, indicating that there were some differences in community diversity among 35 barley rhizosphere soil samples, but the difference was not obvious (Table 5).

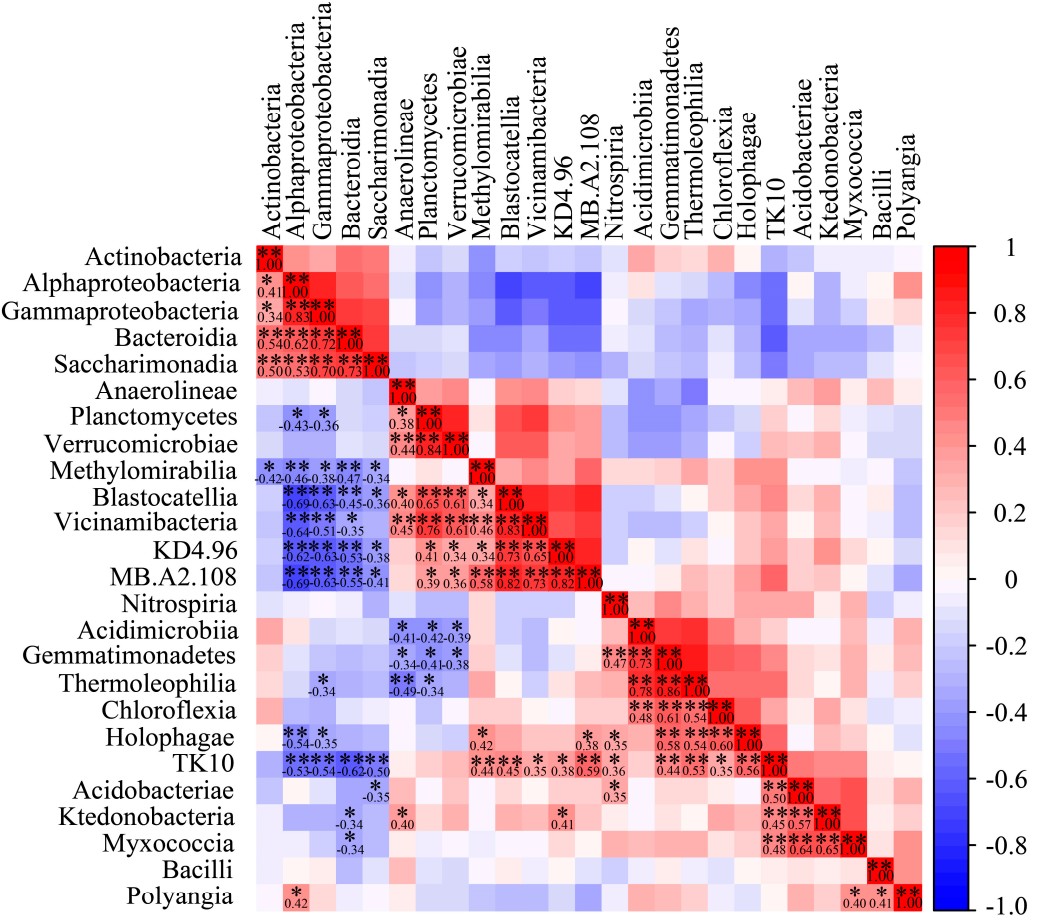

**Figure 5.** Correlation of dominant bacterial communities in barley rhizosphere at class level. * and ** indicate significant correlation at 0.05 and 0.01 levels, respectively.

**Table 5.** Overview of bacterial diversity index in barley rhizosphere soil.

|  | **Minimum** | **Maximum** | **Mean Value** | **Standard Deviation** | **Coefficient of Variation** |
|---|---|---|---|---|---|
| sobs | 2062.00 | 2534.00 | 2273.9143 | 128.6018 | 0.0566 |
| Shannon | 6.31 | 6.62 | 6.4766 | 0.0877 | 0.0135 |
| Simpson | 0.00 | 0.01 | 0.0040 | 0.0008 | 0.1925 |
| ace | 2766.30 | 3394.67 | 3033.6982 | 154.3249 | 0.0509 |
| chao | 2731.20 | 3427.66 | 3051.9101 | 154.8281 | 0.0507 |
| coverage | 0.97 | 0.98 | 0.9735 | 0.0022 | 0.0022 |

### 3.7. Beta Diversity of Bacterial Community Structure in Barley Rhizosphere

In order to compare the differences and similarities of bacterial community structure in rhizosphere soil from 35 different barley varieties, 35 barley rhizosphere soil samples were clustered based on the OTU level of bacterial community. Besides the barley rhizosphere soil sample R32, the OTU level of 35 barley rhizosphere soil samples' bacterial community was mainly clustered into three groups (II, III, and IV). Group II contained seven samples, group III contained fourteen samples, and group IV contained thirteen samples (Figure 6). Principal component analysis was performed on the bacterial composition of 35 barley rhizosphere soil samples. The total interpretation of PCA for the PC1 axis (19.32%) and PC2 axis (18.19%) was 37.51% (Figure 7). The results also showed that the bacteria of 35 samples were mainly clustered into three categories, which was similar to the cluster analysis results (Figure 7).

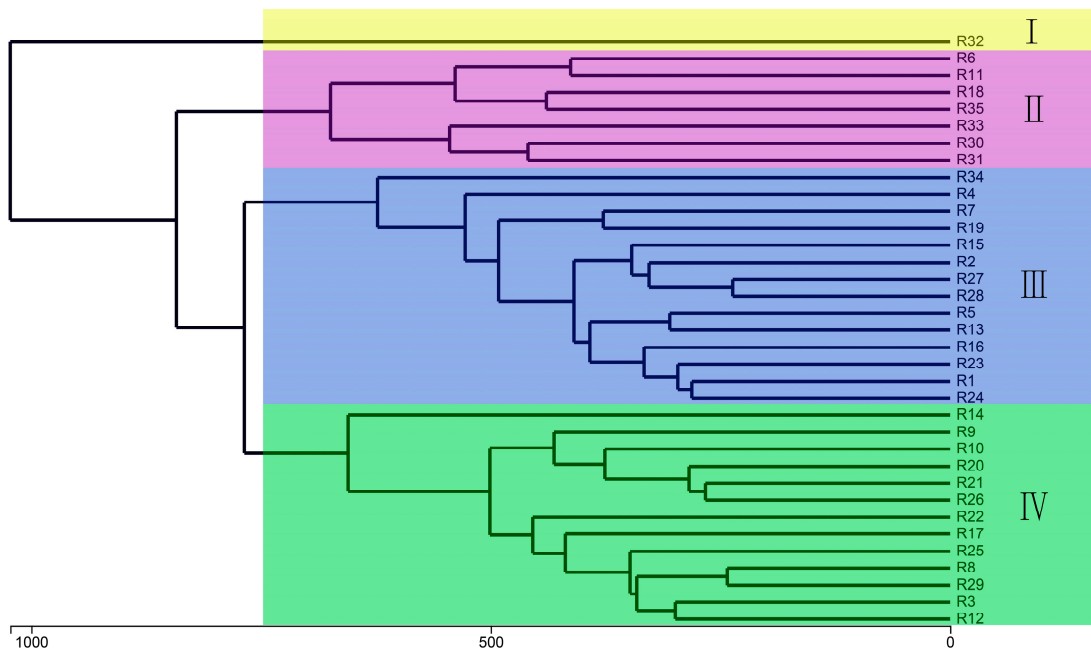

**Figure 6.** Hierarchical clustering of samples at bacterial community phylum level. II, III, and IV indicate the three clusters of rhizosphere soil samples. I indicate the sample that is not included in the three clusters.

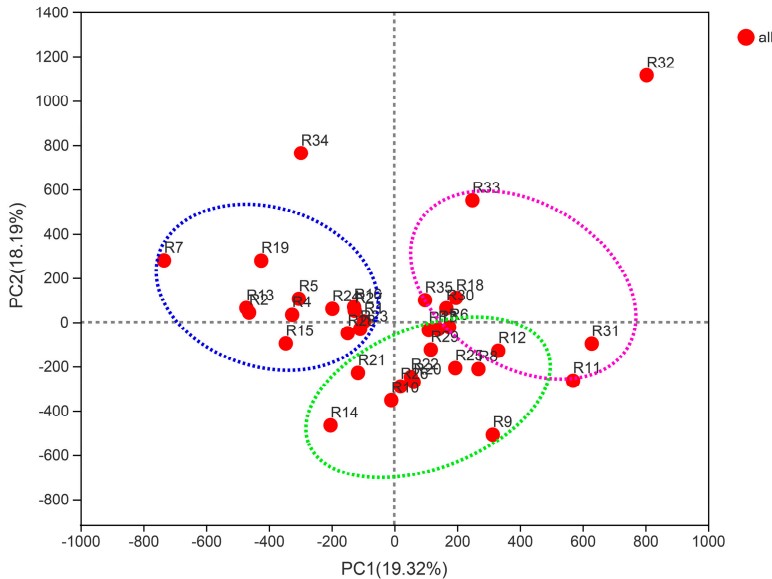

**Figure 7.** PCA analysis of bacterial community phylum level in barley rhizosphere. The purple, blue, and green dashed circles represent groups II, III, and IV, respectively, shown in Figure 6.

In order to detect whether the differences were significant among the three clusters obtained from the beta diversity analysis at the phylum level, we conducted permutational multivariate analysis of variance (PERMANOVA) by 1000 bootstraps (Table 6). As a result, the model explained 29.79% of variation. In addition, both the *p*-value and adjusted *p*-value were lower than 0.01, suggesting that the three clusters exhibited significant differences at the phylum level. The five alpha diversity indexes of the three clusters are listed in Table S4. As a result, clusters II and IV showed lower values than III in sobs, Shannon, ace, and chao indexes, while they showed higher values than III in the Simpson index. These findings suggested that clusters II and IV have lower alpha diversity than III.

**Table 6.** Overview of PERMANOVA.

| Total Variance | Average Variance | F Model | R$^2$ | $p$ | P$_{adjust}$ |
|---|---|---|---|---|---|
| 0.08966 | 0.04483 | 6.57782 | 0.29794 | 0.00813 | 0.0092 |

Note: F Model: F-test value; R$^2$: the explanatory variation by clustering; $p$: significance test for clustering; P$_{adjust}$: adjusted $p$-value by 1000 bootstraps.

### 3.8. Differences in Rhizosphere Bacterial Communities of Different Barley Genotypes

The bacterial community in the rhizosphere of barley had different performances in different genotypes. We compared the abundance of dominant bacterial communities in the three main groups II, III, and IV of cluster analysis and PCA analysis. The five most abundant phyla were Actinobacteriota, Proteobacteria, Acidobacteriota, Chloroflexi, and Gemmatimonadota. According to Figure 8, Proteobacteria and Chloroflexi were significantly different ($p < 0.01$) among different groups. Figure 8A is a schematic diagram of the difference in the abundance of Proteobacteria among the three groups. At the 95% confidence interval, we can see that the average abundance of Proteobacteria in groups III and IV was significantly higher than that in group II, while there was no significant difference between groups III and IV in Proteobacteria. The plot in Figure 8B is a schematic diagram of the difference in the abundance of Chloroflexi among the three groups. At the 95% confidence interval, the average abundance of group II was significantly higher than the average abundance of Chloroflexi in groups III and IV, and there was no significant difference between groups III and IV.

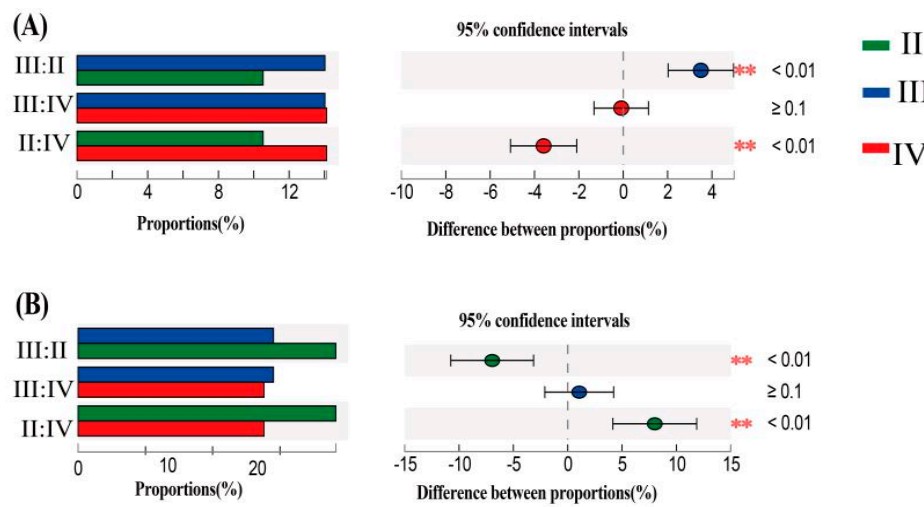

**Figure 8.** Community distribution of different genotypes of barley. II, III, and IV indicate the three clusters of rhizosphere soil samples shown in Figure 6. (**A**). Difference comparison among the three clusters in Proteobacteria. (**B**). Difference comparison among the three clusters in Chloroflexi. ** indicates significant difference between two clusters.

## 4. Discussion

The rhizosphere is an important site for the interaction between plants and rhizosphere microorganisms [25], which can regulate plant nutrient uptake and protect the plant from biotic and abiotic stresses [26,27], thereby promoting the sustainable growth and development of crops. In turn, plants provide root exudates as growth substrates and signal molecules for rhizosphere microorganisms [28]. Crop type is an important factor affecting the composition of rhizosphere bacterial communities [29]. In this study, the composition and structure of rhizosphere bacterial communities in the soil from 35 barley varieties at maturity were analyzed. The results showed significant differences in rhizosphere bacterial communities among 35 rhizosphere soil samples. Among the most abundant phyla, Proteobacteria had higher abundance in clusters III and IV (Figure 8A), while Chloroflexi had

higher abundance in cluster II (Figure 8B). Proteobacteria can promote plant growth and health through various mechanisms, including providing the necessary nutrients for plants, producing plant hormones, inhibiting plant pathogens, and inducing plant resistance to stress [30]. Chloroflexi in soil has the ability to fix nitrogen and dissolve phosphorus [31]. A total of thirteen dominant phyla (relative abundance > 1%) were detected, of which the three most abundant phyla were Actinobacteriota, Proteobacteria, and Acidobacteriota, which was consistent with the result from the rhizosphere soil of *Caragana korshinskii* in cold and arid areas [32], and sorghum, millet, and millet grain [33]. At the phylum level of the 35 samples, R33 (Edamai 720135) showed higher abundance of Proteobacteria, while R32 (Edamai 523898) showed the lowest abundance of Actinobacteriota (Figure 2). This suggests that the two accessions exhibited different effects on the two most abundant phyla. Actinobacteriota has the ability to decompose organic compounds such as plant residues, cellulose, and lignin [34]. Therefore, our subsequent research should focus on the differences in Actinobacteriota between the two varieties and to seek the impact of the difference on barley phenotypes. Twelve dominant bacterial communities (relative abundance > 1%) in the rhizosphere soil of rice and wheat were observed, among which the top three were Proteobacteria, Bacteroidetes, and Acidobacteriota, accounting for more than 50% of the bacterial communities in the rhizosphere soil of rice and wheat [35]. These results suggest that different crop types cause the difference in rhizosphere bacterial community composition. The formation of the rhizosphere microbial community is the result of the joint selection of plant and environmental factors, indicating that crop genotypes have an important influence on the composition of the rhizosphere microbial community. Similar dominant bacteria were observed in cold-resistant and drought-resistant crops such as barley, sorghum, and *Caragana korshinskii*. Actinomycetes and Proteobacteria were confirmed to have a positive effect on the improvement of plant resistance [36], and they might become targeted phyla to study the drought tolerance phenotypes and mechanisms of plants. However, in the current research, our focus was not on the association between plant phenotype and bacterial communities. In addition to uniform soil fertility before sowing (Table 1), the abundance of original bacteria should be a necessary control condition. However, in actual environmental conditions, it is impossible to maintain a uniform abundance of bacterial communities in an experimental field. In future research, it is necessary to adopt a more appropriate approach to study the impact of Actinomycetes and Proteobacteria on barley phenotype.

In this study, the dominant bacterial species in the 35 barley rhizosphere soil samples were basically the same, but differences in abundance were observed. Since 35 different barley varieties were planted in the experimental field of Huazhong Agricultural University, there was no significant difference in the chemical properties and geographical location of the soil. The bacterial community structure and composition in the soil from different barley varieties planted in the same latitude and longitude were similar, but the difference in abundance of bacterial community composition can still be observed from different classification levels of bacteria. This is consistent with a previous suggestion that the effect of planting area on plant rhizosphere microbial community is greater than that of crop varieties [15]. These mutually verifiable conclusions from a previous study [15] and our study here can guide research on the adaptability of barley to extremely harsh soil environments. Only 25 of 35 phylums observed in 35 barley rhizosphere soil samples were shared by 35 samples, and 15 phyla were unique to the rhizosphere from some barley varieties. Dadabacteria only existed in the rhizosphere soil samples from two barley varieties, which was the least covered sample among all bacteria and belonged to a special bacterial phylum. Some bacterial phyla that have a certain effect on plant growth reported in previous studies were also observed in lower abundance, among which the more typical was Deinococcus-Thermus (Deinococcota) [37]. This bacterium has the characteristics of resisting the lethal effects of ionizing radiation and ultraviolet radiation [38]. At the same time, this bacterial phylum is widely present in species that grow at high temperatures. A total of 14 of the 35 samples contain Deinococcus-Thermus (Deinococcota). In summary, it

shows that the composition of rhizosphere bacterial communities among different barley varieties was specific.

## 5. Conclusions

In this study, we investigated bacterial communities in the rhizosphere soil of 35 different barley varieties at the phylum, class, order, and family levels. The findings revealed that Actinobacteriota, Proteobacteria, Acidobacteriota, Chloroflexi, and Gemmatimonadota were the most abundant bacterial communities in the barley rhizosphere at the phylum level. Although the abundance of bacteria among varieties differed relatively little, the abundance of the same bacteria in the rhizospheres of different varieties was different. We identified three main clusters based on the abundance of bacterial communities and found significant differences in bacterial communities of Proteobacteria and Chloroflexi among the sample clusters. Our study provides valuable information for understanding the interaction between rhizosphere microorganisms and crop varieties.

**Supplementary Materials:** The following supporting information can be downloaded at: https://www.mdpi.com/article/10.3390/agronomy13112825/s1, Figure S1: Community abundance at phylum level; Figure S2: Abundance at the order level of barley rhizosphere bacteria; Figure S3: Abundance at the family level of barley rhizosphere bacteria; Table S1: Valid sequences and lengths of the 35 samples; Table S2: Overview of percentage data of dominant bacteria at order level; Table S3: Overview of percentage data of dominant bacteria at family level; Table S4: Six alpha diversity indexes of the three clusters.

**Author Contributions:** Conceptualization, X.R.; methodology, S.Z. and Y.A.; software, Y.Z. and Y.T.; validation, X.W. and D.Z.; formal analysis, Y.A.; investigation, S.Z.; resources, X.W.; data curation, Y.A. and Q.W.; writing—original draft preparation, S.Z. and Y.Z.; writing—review and editing, G.S.; visualization, Y.A.; supervision, X.R.; project administration, X.R.; funding acquisition, X.R. All authors have read and agreed to the published version of the manuscript.

**Funding:** This work was supported by the earmarked fund for CARS (CARS-5), the National Natural Science Foundation of China (32001493).

**Data Availability Statement:** The raw data presented in this study have been deposited in NCBI under the BioProject PRJNA1026781.

**Conflicts of Interest:** The authors declare no conflict of interest.

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
