# Peer review of "Bacterial Community Structure in Rhizosphere of Barley at Maturity Stage"

_agronomy, doi:10.3390/agronomy13112825_

Round 1
Reviewer 1 Report
Comments and Suggestions for Authors
The current manuscript "Bacterial community structure in rhizosphere of barley at maturity stage" The idea seems to be good and the results are promising for publication. However, some comments and modification have to address before publication.
1. The abstract part needs to rewrite in a way to define the exact novelty and originality of your work.
2. All abbreviations used should be mentioned in the place of their first mention followed by an abbreviation and then only the abbreviation is written.
3. The introduction must be completed by clarifying the main objectives of the research and by motivating the experimental strategy adopted by authors.
4. The introduction contains many paragraphs without references, such as lines 45-61. Authors must add references to these paragraphs
5. Add all materials used in Materials section.
6. Add references to all methods.
7. The authors should add more explanations to the results and discussion.
8. The authors need to improve the resolution of all figures.
9. The conclusion is poorly; I think author should try to link better their work.
- The whole manuscript must be checked to avoid the presentation of same information several times.
11. The text must be checked carefully in order to correct the editing errors.
Comments on the Quality of English Language
- The English language needs to be improved in wording, grammar, and sentence structure.
Author Response
The current manuscript "Bacterial community structure in rhizosphere of barley at maturity stage" The idea seems to be good and the results are promising for publication. However, some comments and modification have to address before publication.
Response: Thank you for your recognition of our work. We have made revisions to the manuscript based on your comments, and have responded to your comments point by point. We hope our revised version meets the standards for publication.
- The abstract part needs to rewrite in a way to define the exact novelty and originality of your work.
Response: Thank you. We have re-organized the abstract by exacting the main results. We think the revised version is more prominent in novelty and originality.
- All abbreviations used should be mentioned in the place of their first mention followed by an abbreviation and then only the abbreviation is written.
Response: Thank you. We have checked it.
- The introduction must be completed by clarifying the main objectives of the research and by motivating the experimental strategy adopted by authors.
Response: We have revised it. we examine the rhizosphere bacterial community structure in the mature stage of 35 barley varieties by 16S rDNA gene sequencing. The goal was to determine the dominant bacterial communities present in the rhizosphere of barley, as well as to assess the diversity and differences among the different varieties.
- The introduction contains many paragraphs without references, such as lines 45-61. Authors must add references to these paragraphs.
Response: We have added references to these paragraphs.
- Add all materials used in Materials section.
Response: Thank you for your suggestion. We have added a table that list the 35 barley varieties in Materials section.
- Add references to all methods.
Response: We have added.
- The authors should add more explanations to the results and discussion.
Response: Thank you. We have re-organized the discussion section and further discussed some of the results.
- The authors need to improve the resolution of all figures.
Response: We have improved our resolution of all figures.
- The conclusion is poorly; I think author should try to link better their work.
Response: Thank you for your suggestion. We have improved our conclusion.
- The whole manuscript must be checked to avoid the presentation of same information several times.
Response: Thank you. We have checked our manuscript carefully.
- The text must be checked carefully in order to correct the editing errors.
Response: Thank you. We have checked it.
Reviewer 2 Report
Comments and Suggestions for Authors
There are no questions about the methodology for studying the bacterial community. However, there are questions about setting up the experiment. The conclusions and results lack specificity.
1) This is not the first study of the barley rhizosphere. In particular, the authors themselves refer to works on eco-geographical patterns in the bacterial community of the barley rhizosphere. What new did the authors manage to show?
2) From the results it follows that the rhizosphere bacterial community of barley is generally similar, which is expected. Were unique phyla somehow related to the properties of barley varieties? The article makes no attempt to explain the differences between varieties. The properties of the varieties are not given.
3) All varieties were grown under the same conditions. Aren't these varieties designed to grow in certain natural areas? Apparently, for some varieties the conditions may not have been optimal? What is the point of this approach?
4) The discussion section is very small and needs to be expanded (see above).
Author Response
There are no questions about the methodology for studying the bacterial community. However, there are questions about setting up the experiment. The conclusions and results lack specificity.
Response: Thank you for your valuable comments and suggestions on our manuscript. We have reorganized the results and conclusions based on suggestions from you and other reviewers.
1)This is not the first study of the barley rhizosphere. In particular, the authors themselves refer to works on eco-geographical patterns in the bacterial community of the barley rhizosphere. What new did the authors manage to show?
Response: Thank you. In order to study the geographical ecological patterns of rhizosphere microbial communities in different ecological regions of barley, we should plant these 35 varieties in different ecological regions of barley, and take rhizosphere samples at the same growth period, and record environmental factors (such as temperature, humidity, soil physical and chemical properties) and barley phenotypes in different ecological regions. We manage to show: 1) Does the same variety have consistency in shaping microbial communities in different ecological regions? 2) How environmental factors shape microbial communities?
2)From the results it follows that the rhizosphere bacterial community of barley is generally similar, which is expected. Were unique phyla somehow related to the properties of barley varieties? The article makes no attempt to explain the differences between varieties. The properties of the varieties are not given.
Response: Thank you. According to your suggestion, we have added this in results and discussion section: “In the phyla level of the 35 samples, R33 (Edamai 720135) showed higher abundance of Proteobacteria, while R32 (Edamai 523898) showed the lowest abundance of Actinobacteriota (Figure 2). This suggests that the two accessions exhibit different effects on the two most abundant phyla. Actinobacteriota has the ability to decompose organic compounds such as plant residues, cellulose, and lignin. Therefore, our subsequent research should focus on the differences in Actinobacteriota between the two accessions and to seek the impact of the difference on barley phenotypes.”
3)All varieties were grown under the same conditions. Aren't these varieties designed to grow in certain natural areas? Apparently, for some varieties the conditions may not have been optimal? What is the point of this approach?
Response: Thank you. the purpose of this manuscript is to investigate the structure and diversity of bacterial communities in the rhizosphere of different barley varieties. Under the uniform fertility status of the experimental plot. We want to know whether the abundances of bacteria among varieties have large variation. In addition, we also want to find some bacteria that play a dominant role in certain varieties. Actually, for some varieties the conditions may not have been optimal. However, our method can more clearly observe the impact of rhizosphere microbial communities on different barley varieties.
4)The discussion section is very small and needs to be expanded (see above).
Response: Thank you. We have expanded the discussion section based on suggestions from you and other reviewers.
Reviewer 3 Report
Comments and Suggestions for Authors
The study by Zhang et al investigates microbial composition in rhizosphere of 35 barley varieties applying next-generation sequencing of 16S rRNA gene fragments.
The work is mostly descriptive and does not display significant scientific findings. The authors are engaged in endless enumeration of bacterial taxa at all taxonomic ranks. However, no assumptions are made about the reason why this or that taxon would predominate. Or which bacterial groups distinguish the barley rhizosphere from the rhizosphere of other crops and why.
The methodology section is not clearly written. The authors write about 35 crop varieties that grew each in 3 rows and in 3 replicates (we can assume that there were 3 biological and 3 technical replicates for each variety). However, only 35 samples were used for DNA isolation. At what stage then were the replicates involved?
It is not clear how the correlation analysis was performed in the case of a single sample per barley variety. Correlation analysis requires replicates.
The Discussion section should contain more comparisons with data from other papers and not just numerical ones. There should be a scientific concept and an idea in such comparisons.
The Conclusion section should not contain a numeric list of results, but a generalization of the research, what the study has contributed to this area of science.
The overall impression is that the paper is rather weak to be published in this journal. If the authors can correct many conceptual flaws, the paper could be considered for review again.
Minor comments:
Lines 65-70 – Rephrase
Line 87-88 double “root”
Section 2.4 Change “Statistical analyses” to “Bioinformatical analyses”
Line 101-109: misspelling OTU
Line 101-109: Please, add some information about percentage that was used for taxonomy assignment (now only for OTU clustering)
Line 101-109: Add info about software that was used for Pearson test
Line111: Add some sequencing statistics (number of raw reads and quality filtered reads, number of merged paired reads)
Line 154-160: repharse
Line 164: not 92 dominant classes, but minor (if their relative abundance<1%)
Line 323: can still be observed on different taxonomy levels of bacteria
Line 326-329: repeat of results lines, should be discussion about this finding
Line 334-336: why do you find deinococcus thermus. Current discussion is not applicable to your environment (both temperature and radiation part). Again, the discussion should be consistent with your results and overall experiment
Author Response
The study by Zhang et al investigates microbial composition in rhizosphere of 35 barley varieties applying next-generation sequencing of 16S rRNA gene fragments.
The work is mostly descriptive and does not display significant scientific findings. The authors are engaged in endless enumeration of bacterial taxa at all taxonomic ranks. However, no assumptions are made about the reason why this or that taxon would predominate. Or which bacterial groups distinguish the barley rhizosphere from the rhizosphere of other crops and why.
Response: Thank you for giving us a chance to revise our manuscript. Your comments are very valuable. In this work, we showed bacterial taxa at all taxonomic ranks to investigate the structure and diversity of bacterial communities in the rhizosphere of different barley varieties. In the revised version, we made assumptions about the reason why this or that taxon would predominate, and discussed it in the discussion section. We also compared the dominant bacterial groups in rhizosphere between barley and other crops in the discussion section.
The methodology section is not clearly written. The authors write about 35 crop varieties that grew each in 3 rows and in 3 replicates (we can assume that there were 3 biological and 3 technical replicates for each variety). However, only 35 samples were used for DNA isolation. At what stage then were the replicates involved?
Response: Thank you. We planted the 35 varieties with 3 replicates (each replicate with 3 rows). At the maturity stage of barley, we took three samples from each replicate to obtain 9 soil samples for each variety. Then we cut the root of a 2-7cm depth of these 9 samples of each variety and collected the rhizosphere soil. In order to ensure sufficient rhizosphere soil for sequencing, we mixed the 9 samples of each variety. Finally, we obtained 35 samples for next-generation sequencing of 16S rRNA gene fragments. We have made the methodology section clearer in the revised version.
It is not clear how the correlation analysis was performed in the case of a single sample per barley variety. Correlation analysis requires replicates.
Response: Correlation heatmaps of figure 4 and figure 5 were drawn by the R package corrplot. Correlation analyses were performed by the abundance of predominate bacterial communities of the 35 samples at phylum and class levels,respectively.
The Discussion section should contain more comparisons with data from other papers and not just numerical ones. There should be a scientific concept and an idea in such comparisons.
Response: Thank you. We have improved our manuscript in the Discussion section by comparing our results with that in sorghum, millet, wheat and rice.
The Conclusion section should not contain a numeric list of results, but a generalization of the research, what the study has contributed to this area of science.
Response: Thank you, we have revised it.
The overall impression is that the paper is rather weak to be published in this journal. If the authors can correct many conceptual flaws, the paper could be considered for review again.
Response: Thank you. We are making every effort to improve our manuscript to meet the standards for publication in Agronomy.
Minor comments:
Lines 65-70 – Rephrase
Response: Thank you, we have revised it.
Line 87-88 double “root”
Response: Thank you, we have checked it.
Section 2.4 Change “Statistical analyses” to “Bioinformatical analyses”
Response: Thank you, we have revised it.
Line 101-109: misspelling OTU
Response: Thank you, we have checked it.
Line 101-109: Please, add some information about percentage that was used for taxonomy assignment (now only for OTU clustering)
Response: Thank you, we have added it.
Line 101-109: Add info about software that was used for Pearson test.
Response: Thank you, we have added it.
Line111: Add some sequencing statistics (number of raw reads and quality filtered reads, number of merged paired reads)
Response: Thank you, we have added sequencing statistics in table S1.
Line 154-160: repharse
Response: Thank you, we have revised it.
Line 164: not 92 dominant classes, but minor (if their relative abundance<1%)
Response: Thank you, we have checked it.
Line 323: can still be observed on different taxonomy levels of bacteria
Response: Thank you, we can observe the different taxonomy levels.
Line 326-329: repeat of results lines, should be discussion about this finding
Response: Thank you, we discussed it further.
Line 334-336: why do you find deinococcus thermus. Current discussion is not applicable to your environment (both temperature and radiation part). Again, the discussion should be consistent with your results and overall experiment.
Response: Thank you. The purpose to describe Deinococcus-Thermus here is to show the result and make a further discussion. In order to make consistency between results and discussions, we added the result of the Deinococcus-Thermus to the result section.
Reviewer 4 Report
Comments and Suggestions for Authors
The article is devoted to the study of the structure of the bacterial community in the rhizosphere of barley at the maturity stage. The interaction of plants and soil bacteria in evolutionary terms has provided significant advantages in their development, especially on low-fertility soils. This is very important for agricultural cultures, where the goal is to obtain a high-quality harvest with an increase in its volume. Therefore, the study of the interaction between soil microflora and the root system of plants is relevant. The authors substantiated the role of barley in the economy and food security of Tibet. Using 35 barley varieties as an example, a study was conducted of the composition of the dominant bacterial communities in the barley rhizosphere, as well as differences in the diversity and population structure of rhizosphere microbial communities between varieties. There are questions and comments about the article to improve it.
1. From Table 1 it is not clear how many soil samples were analyzed and statistical parameters are not indicated (average, geometric mean, min., max., etc.).
2. Section 2.2 describes rhizosphere sampling. How was the identity of rhizosphere selection controlled with such manual labor? In some places it was possible to remove more soil with a brush, in others less.
3. Why has not the microbial community of soil without plants been studied, i.e. soil bacterial background? What can be considered control in this study?
4. In the discussion section, it is necessary to more clearly indicate which barley varieties had more species in the rhizosphere and which had a larger quantity, otherwise the result of the work is not obvious for practice. The conclusions also talk about varietal specificity without specifying the names of the varieties.
5. The main question is that only the species composition of the microbiota was studied. Why wasn’t the yield of the barley varieties under study, the quality of the seeds, and why wasn’t the influence of microbiota on barley yield compared?
Author Response
The article is devoted to the study of the structure of the bacterial community in the rhizosphere of barley at the maturity stage. The interaction of plants and soil bacteria in evolutionary terms has provided significant advantages in their development, especially on low-fertility soils. This is very important for agricultural cultures, where the goal is to obtain a high-quality harvest with an increase in its volume. Therefore, the study of the interaction between soil microflora and the root system of plants is relevant. The authors substantiated the role of barley in the economy and food security of Tibet. Using 35 barley varieties as an example, a study was conducted of the composition of the dominant bacterial communities in the barley rhizosphere, as well as differences in the diversity and population structure of rhizosphere microbial communities between varieties. There are questions and comments about the article to improve it.
Response: Thank you for your valuable comments and suggestions on our manuscript. We greatly appreciate your input as it is highly beneficial in enhancing the quality of our work. We have carefully considered all your suggestions and have made revisions to improve the technical soundness of our manuscript.
1. From Table 1 it is not clear how many soil samples were analyzed and statistical parameters are not indicated (average, geometric mean, min., max., etc.).
Response: Thank you, Table 1 shows average level of the seven soil chemical properties in the experimental plot before sowing. It describes the uniform fertility status of the experimental plot, and the data were gained from ten randomly selected sites in the experimental field. The original fertility status of the experimental field is not the focus of the research, so we only listed the average level of the seven soil chemical properties.
2. Section 2.2 describes rhizosphere sampling. How was the identity of rhizosphere selection controlled with such manual labor? In some places it was possible to remove more soil with a brush, in others less.
Response: Thank you, this is a very valuable question. Actually, we collected the soil profiles of 2-7 cm depth from each barley accession and cut the roots (includes main and fine roots) of 2-5 cm depth for obtaining enough rhizosphere soil samples. Barley roots in 8-15 cm depth are not considered by us because they are not sufficient for collection of rhizosphere soil samples. Moreover, they might show significant heterogeneity with roots of 5-15 cm depth in the structure and diversity of the bacterial community. We revised it in our manuscript.
3.Why has not the microbial community of soil without plants been studied, i.e. soil bacterial background? What can be considered control in this study?
Response: Thank you, the purpose of this manuscript is to investigate the structure and diversity of bacterial communities in the rhizosphere of different barley varieties. Under the uniform fertility status of the experimental plot. We want to know whether the abundances of bacteria among varieties have large variation. In addition, we also want to find some bacteria that play a dominant role in certain varieties. The 35 varieties were planted in the same type of soil, which is uniform for all varieties. If the 35 varieties were planted in different locations, it is very important to measure the microbial community of soil without plants been studied. Therefore, our results still provide a reference for plant microbial interactions. We have discussed this in our manuscript.
4.In the discussion section, it is necessary to more clearly indicate which barley varieties had more species in the rhizosphere and which had a larger quantity, otherwise the result of the work is not obvious for practice. The conclusions also talk about varietal specificity without specifying the names of the varieties.
Response: Thank you. According to your suggestion, we have added this in results and discussion section: “In the phyla level of the 35 samples, R33 (Edamai 720135) showed higher abundance of Proteobacteria, while R32 (Edamai 523898) showed the lowest abundance of Actinobacteriota (Figure 2). This suggests that the two accessions exhibit different effects on the two most abundant phyla. Actinobacteriota has the ability to decompose organic compounds such as plant residues, cellulose, and lignin. Therefore, our subsequent research should focus on the differences in Actinobacteriota between the two accessions and to seek the impact of the difference on barley phenotypes.”
5.The main question is that only the species composition of the microbiota was studied. Why wasn’t the yield of the barley varieties under study, the quality of the seeds, and why wasn’t the influence of microbiota on barley yield compared?
Response: Thank you. This is a very valuable question. The purpose of this study is to investigate the structure and diversity of bacterial communities in the rhizosphere of different barley varieties. The abundance of bacteria among varieties differed relatively low (i.e., the dominant microbial communities are consistent across varieties), but the abundance of the same bacteria in the root soil of different varieties was different (such as Actinomycetes and Proteobacteria). The lack of association with phenotype is a limitation in this manuscript, which we presented in our discussion. This will be focused in our future work.
Round 2
Reviewer 3 Report
Comments and Suggestions for Authors
The authors have addressed the previous comments. There are repeated words in the Conclusion, try to reword it, it will improve the readers' perception of the text.
Author Response
The authors have addressed the previous comments. There are repeated words in the Conclusion, try to reword it, it will improve the readers' perception of the text.
Response: Thank you for your recognition of our work. Based on your suggestion, we have revised the conclusion section. We hope our revisions can make the article more readable.
Reviewer 4 Report
Comments and Suggestions for Authors
The authors significantly revised the manuscript, corrected all the reviewer’s comments or gave a substantiated answer to why the study was conducted in such a design. I think that the article has improved significantly and can be accepted for publication.
Author Response
The authors significantly revised the manuscript, corrected all the reviewer’s comments or gave a substantiated answer to why the study was conducted in such a design. I think that the article has improved significantly and can be accepted for publication.
Response: Thank you for your efforts in improving this manuscript. In order to make the manuscript more readable, we made substantial revisions in the first round of revision (include a clearer presentation of the experimental design). In the second round of revision, we reworded the conclusion section. We hope this manuscript will receive more attentions after publication.